# Bafilomycin A1 Inhibits HIV-1 Infection by Disrupting Lysosomal Cholesterol Transport

**DOI:** 10.3390/v16091374

**Published:** 2024-08-29

**Authors:** Byeongwoon Song, Olga Korolkova

**Affiliations:** Department of Microbiology, Immunology and Physiology, Center for AIDS Health Disparities Research, School of Medicine, Meharry Medical College, Nashville, TN 37208, USA; okorolkova@mmc.edu

**Keywords:** HIV-1, lysosome, bafilomycin A1, cholesterol transport, NPC1

## Abstract

The productive replication of human immunodeficiency virus type 1 (HIV-1) involves intricate interactions between viral proteins and host cell machinery. However, the contributions of the lysosomal pathways for HIV-1 replication are not fully understood. The goal of this study was to determine the impact of lysosome-targeting compounds on HIV-1 replication and identify the cellular changes that are linked to HIV-1 inhibition using cell culture models of HIV-1 infection. Here, we demonstrate that the treatment of cells with various pharmacological agents known to inhibit lysosomal functions interfere with HIV-1 replication. The vacuolar ATPase (V-ATPase) inhibitor bafilomycin A1 exerted a potent inhibition of HIV-1 replication. Bafilomycin A1 inhibition of HIV-1 was independent of coreceptor tropism of HIV-1. Our data suggest that bafilomycin A1 inhibits HIV-1 at the post-integration steps of the virus life cycle, which include viral gene expression, virus assembly, and/or egress. Analysis of the cellular alterations following bafilomycin A1 treatment indicates that bafilomycin A1 causes a disruption in lysosome structure and functions. Treatment of cells with bafilomycin A1 caused an accumulation of unesterified cholesterol in lysosomes along with the expansion of the lysosomal compartments. Interestingly, the overexpression of the lysosomal cholesterol transporter Niemann–Pick type C 1 (NPC1) partially relieved bafilomycin A1 inhibition of HIV-1. Collectively, our data suggest that bafilomycin A1 inhibits HIV-1 replication in part by disrupting the lysosomal cholesterol trafficking pathway.

## 1. Introduction

Human immunodeficiency virus (HIV-1), a causative agent for AIDS, is an enveloped retrovirus with a single-stranded RNA genome that requires complex interactions with the host cell machinery to complete the viral life cycle [1]. The global AIDS pandemic continues despite significant efforts to curb the spread of HIV-1. More than 39 million people are living with HIV-1 worldwide and approximately 1.3 million more people are newly infected with HIV-1 each year [2]. Currently, there is no vaccine available for preventing HIV-1 infection. There are several antiretroviral drugs that can inhibit HIV-1 replication and delay the onset of AIDS. However, the emergence of viral resistance as well as the adverse side effects and high costs of these drugs are serious issues that warrant further research. Therefore, there is a compelling need to better understand the virus‒host cell interactions that can facilitate the development of improved therapeutic strategies. Despite the steady progress in our knowledge of HIV-1 biology, there is a significant gap in our understanding of the role of the lysosomal pathway in the HIV-1 replication cycle. 

Lysosomes (here, the term ‘lysosome’ collectively refers to late endosomes, lysosomes, and endolysosomes) are membrane-enclosed structures that are crucial for cellular health and energy homeostasis. Cholesterol plays a critical role throughout the HIV-1 life cycle [3,4,5,6] viral entry, assembly, and budding occur at the cholesterol-enriched membrane microdomains known as lipid rafts. Importantly, lysosomes play a crucial role in maintaining cellular cholesterol homeostasis [7]. The endosomal/lysosomal pathway is also known to be involved in the late stages of HIV-1 replication, including virus assembly and budding [8,9,10,11,12]. Given the role of lysosomes in cellular cholesterol homeostasis and the need of HIV-1 for cholesterol, it is highly likely that a disruption in the lysosomal processes could affect HIV-1 replication. 

Lysosome-targeting compounds have been invaluable for elucidating the interplay between lysosomal processes and the viral life cycle. Some lysosome-targeting compounds have been tested for their potential to treat viral infections, especially diseases caused by viruses that rely on the endocytic pathway for cell entry [13,14]. However, the impact of lysosome inhibition on HIV-1 replication is not fully understood. Furthermore, discrepancies exist among studies that have examined the effects of lysosome-targeting compounds on HIV-1 replication [15,16,17,18,19]. To address these discrepancies and to understand the functional significance of lysosomal pathway in the HIV-1 life cycle, we tested the effects of three lysosome-targeting compounds on HIV-1 infection. Of the compounds tested, the vacuolar ATPase (V-ATPase) inhibitor bafilomycin A1 [20] showed the most potent antiviral activity. Further dissecting the changes in cellular pathways and functions uncovered upon bafilomycin A1 treatment could provide important new insights for understanding virus–host cell interactions and designing strategies to block HIV-1 replication.

## 2. Materials and Methods

### 2.1. Reagents and Antibodies

We obtained the reagents used in cell culture from Thermo Fisher Scientific (Waltham, MA, USA), including Dulbecco’s modified Eagle’s medium (DMEM), RPMI 1640 medium, fetal bovine serum (FBS), penicillin, and streptomycin. The JetPrime transfection reagent was obtained from VWR (Radnor, PA, USA). The CellTiter 96 Aqueous One Solution Cell Proliferation Assay System (MTS assay) and Luciferase Assay System were purchased from Promega (Madison, WI, USA). The QIAamp DNA Blood Mini Kit was obtained from Qiagen (Hilden, Germany). Deoxyribonucleotide triphosphate (dNTP) mix was purchased from Thermo Fisher Scientific (Waltham, MA, USA). The Hot Start Taq DNA polymerase and the standard Taq reaction buffer were obtained from New England Biolabs (Ipswich, MA, USA). The SsoAdvanced Universal SYBR green supermix was purchased from Bio-Rad Laboratories (Hercules, CA, USA). Bafilomycin A1, chloroquine, and U18666A were purchased from Sigma-Aldrich (St. Louis, MO, USA). The antiretroviral drugs T20, zidovudine, and raltegravir were obtained from the NIH HIV Reagent Program. Filipin was obtained from Sigma-Aldrich (St. Louis, MO, USA). The mammalian protein extraction buffer was obtained from VWR (Radnor, PA, USA). The SuperSignal WestDura Extended Duration Substrate was purchased from Thermo Fisher Scientific (Waltham, MA, USA). 

Antibodies for LAMP1 (#10665, #9091) and GAPDH (#5174) are from Cell Signaling Technology (Danvers, MA, USA). Antibodies for TOMM20 (ab78547), calreticulin (ab92516), and GM130 (ab52649) were obtained from Abcam (Cambridge, MA, USA). Antibodies for GFP (sc-9996) and NPC1 (sc-271335) were purchased from Santa Cruz Biotechnology (Dallas, TX, USA). Horseradish peroxidase (HRP)-conjugated anti-mouse (HAF018) and anti-rabbit (HAF008) are from R&D systems (Minneapolis, MN, USA). CF488A-conjugated goat anti-mouse (#20018) and anti-rabbit (#20012) and CF594-conjugated goat anti-mouse (#20110) and anti-rabbit (#20153) were obtained from Biotium (Fremont, CA, USA). 

### 2.2. Cells, Plasmids, and Viruses

The human Jurkat T cell line and the human embryo kidney 293T (HEK293T) cell line were obtained from the American Type Culture Collection (Manassas, VA, USA). HeLa-derived cell line TZM-bl (21) expressing the HIV-1 receptor/coreceptor and HIV-1 LTR-driven luciferase was obtained from the NIH HIV Reagent Program. Jurkat cells were cultured in RPMI 1640 supplemented with 10% heat-inactivated FBS, 100 U/mL of penicillin, and 100 µg/mL of streptomycin at 37 °C in a 5% CO_2_ humidified incubator. HEK293T and TZM-bl cells were cultured in DMEM supplemented with 10% heat-inactivated FBS, 100 U/mL of penicillin, and 100 µg/mL of streptomycin at 37 °C in a 5% CO_2_ humidified incubator. 

The plasmid containing the HIV-1 2-LTR circle (pG-HIV-2LTR; #104590) and the plasmid expressing NPC1-GFP (NPC1-His6-EGFP, #53521) were obtained from Addgene (Watertown, MA, USA). The empty control plasmid pcDNA3.1 is from Thermo Fisher Scientific (Waltham, MA, USA). The infectious HIV-1 molecular clones pNL4.3, pNL4.3-BaL, and p89.6 were obtained from the NIH HIV Reagent Program. Virus supernatants were prepared by transient transfection of HEK293T cells, and p24 content was determined using an enzyme-linked immunosorbent assay (ELISA) with a kit purchased from XpressBio (Frederick, MD, USA). 

### 2.3. Cell Viability Assay

We measured the effect of various compounds on cell viability or cytotoxicity using the CellTiter 96 Aqueous One Solution Cell Proliferation Assay Kit (Promega). Briefly, cells were seeded in 96-well plates at a density of 1–2 × 10^4^ cells per well. Cells were allowed to adhere for one day and then treated with the drugs at the indicated concentration. Cells treated with dimethyl sulfoxide (DMSO) at 0.1% (*v*/*v*) were included as a negative control. After 48 h incubation at 37 °C, we determined cell viability following the manufacturer’s instructions. 

### 2.4. Infection Assays

For virus replication, Jurkat cells were pretreated with the drugs for 2 h and then infected with HIV-1_NL4.3_ (4 ng of p24/2 × 10^5^ cells) in the presence of the drugs. The following day, the cells were washed to remove residual virus and cultured in fresh media with the drugs. At 4 days post-infection, the levels of HIV-1 Gag (p24) released in the culture media were determined using a standard ELISA assay (XpressBio). 

For virus infectivity, TZM-bl cells were pretreated with the drugs for 2 h and then infected with HIV-1 (2 ng of p24/2 × 10^5^ cells) in the presence of the drugs. At 48 h post-infection, cell lysates were prepared, and virus infection was measured using the luciferase assay system (Promega).

A time-of-addition assay was conducted to identify the steps of the virus life cycle which can be inhibited by each compound. TZM-bl cells were infected with HIV-1_NL4.3_ (2 ng of p24/2 × 10^5^ cells) with the drugs added at the indicated times post-infection (0 h, 3 h, 6 h, and 9 h). At 48 h post-infection, cell lysates were prepared, and virus infectivity was determined using the luciferase assay system (Promega).

### 2.5. Quantification of Reverse Transcription Products and 2-LTR Circles

We measured reverse transcription products and 2-LTR circles using a qPCR-based strategy. Reverse transcription products were quantified by utilizing a SYBR green-based qPCR. The reaction mixture included 100 ng of total DNA, 1× SsoAdvanced Universal SYBR green supermix (Bio-Rad Laboratories, Hercules, CA, USA), and 300 nM late RT F (5′-TGTGTGCCCGTCTGTTGTGT-3′) and late RT R (5′-GAGTCCTGCGTCGAGAGAGC-3′) primers. The 2-LTR circles were quantified by utilizing a SYBR green-based qPCR. The reaction mixture included 100 ng of total DNA, 1× SsoAdvanced Universal SYBR green supermix, and 300 nM 2-LTR forward (5′-AACTAGGGAACCCACTGCTTAAG-3′) and 2-LTR reverse (5′-TCCACAGATCAAGGATATCTTGTC-3′) primers. The qPCR conditions consisted of an initial denaturation at 95 °C for 3 min, followed by 39 cycles of amplification and acquisition at 94 °C for 15 s, 58 °C for 30 s, and 72 °C for 30 s. To quantify the reverse transcription products, we generated a standard curve in parallel under the same conditions using 10-fold serial dilutions of known copy numbers (1 × 10^0^ to 1 × 10^8^) of the HIV-1 molecular clone pNL4.3 plasmid (NIH HIV Reagent Program). Similarly, to calculate the 2-LTR copies, we generated a standard curve using 10-fold serial dilutions of known copy numbers (1 × 10^0^ to 1 × 10^8^) of the pG-HIV-2LTR plasmid (Addgene). We then determined the copy numbers of reverse transcription and 2-LTR circles by plotting the qPCR data against the respective standard curve. The qPCRs were performed in triplicates, and the data were analyzed using CFX Maestro software v2.1 (Bio-Rad Laboratories).

### 2.6. Quantification of HIV-1 Integration

We quantified HIV-1 proviral DNA using a nested PCR method, including two rounds of reactions. The first-round endpoint PCR was conducted using primers designed to amplify only the integration junctions between human Alu repeats and HIV-1 viral DNA but not the unintegrated viral DNA. The second-round qPCR was conducted using primers designed to specifically amplify only the viral LTR from the first-round PCR products. The first-round PCR contained 100 ng of total DNA, 1× standard Taq reaction buffer (New England Biolabs), deoxyribonucleotide triphosphate (dNTP) mix containing 200 μM concentrations of each nucleotide (Thermo Fisher Scientific), 500 nM primers targeting Alu repeat sequence (5′-GCCTCCCAAAGTGCTGGGATTACAG-3′) and HIV-1 Gag sequence (5′-GTTCCTGCTATGTCACTTCC-3′), and 1.25 U of Hot Start Taq DNA polymerase (New England Biolabs) in a 50 μL final volume. The first-round PCR was performed under conditions of an initial incubation at 95 °C for 5 min, followed by 23 cycles of amplification at 94 °C for 30 s, 50 °C for 30 s, and 72 °C for 4 min, with a final incubation at 72 °C for 10 min. The second-round qPCR included one-tenth of the product from the first-round PCR as the template DNA, 1× SsoAdvanced Universal SYBR green supermix (Bio-Rad Laboratories), and 300 nM (each) of the viral LTR-specific primers that target the R region (5′-TCTGGCTAACTAGGGAACCCA-3′) and the U5 region (5′-CTGACTAAAAGGGTCTGAGG-3′). The qPCR was performed under conditions of an initial incubation at 95 °C for 3 min, followed by 39 cycles of amplification and acquisition at 94 °C for 15 s, 58 °C for 30 s, and 72 °C for 30 s. We generated a standard curve in parallel under same conditions using 10-fold serial dilutions of known copy numbers (1 × 10^0^ to 1 × 10^8^) of the HIV-1 molecular clone pNL4.3 plasmid (NIH HIV Reagent Program). The qPCR experiments were performed in triplicate, and the data were analyzed using CFX Maestro software (Bio-Rad Laboratories). The integrated viral DNA copy numbers were calculated by plotting the qPCR data against the standard curve. 

### 2.7. Virus Particle Production Assay

HEK293T cells were plated in 12-well plates and allowed to adhere for one day. The following day, the cells were transfected with the proviral molecular clone pNL4-3 plasmid (1 µg per well) using the JetPrime transfection reagent (VWR) according to the manufacturer’s instructions. At 4 h post-transfection, cells were washed to remove the media containing the transfection complex. The cells were then incubated in fresh media in the absence or presence of bafilomycin A1. At 28 h post-transfection, the levels of HIV-1 Gag (p24) released in the culture media were determined using a standard ELISA assay. 

### 2.8. Intracellular Distribution of Cholesterol

To determine intracellular distribution of cholesterol, we washed the cells three times with PBS and fixed them with 3.7% formaldehyde in PBS for 30 min at room temperature. After washing three times with PBS, cells were stained with filipin (12.5 µg/mL in PBS) for 45 min at room temperature. Cells were washed three times with PBS and images were captured using the Nikon A1R confocal microscope with 60× magnification. The mean arbitrary units of fluorescence ± SD were quantified using NIS-Elements software v 4.0 (Nikon). ROIs were manually defined around the cell boundary and mean fluorescence intensity was measured. 

### 2.9. Immunostaining

We fixed cells with 3.7% formaldehyde in PBS for 30 min and washed them three times with PBS. The cells were then permeabilized and blocked for 30 min in 0.1% Triton X-100 (*v*/*v*)/1% bovine serum albumin (*w*/*v*) in PBS. All steps were conducted at room temperature. Permeabilized cells were incubated with primary antibodies at 4 °C overnight. After three washes with PBS, the cells were incubated with secondary antibodies for 2 h at room temperature followed by three washes in PBS. DAPI (1 µM) was used to stain nuclei. Rabbit anti-LAMP1 (#9091; Cell Signaling Technology), rabbit anti-TOMM20 (ab78547; Abcam), rabbit anti-calreticulin (ab92516; Abcam), and rabbit anti-GM130 (ab52649; Abcam) were used at 1:200. The fluorophore-conjugated secondary antibodies (Biotium) were used at 1:200. Images were captured using the Nikon A1R confocal microscope with 60× magnification. The mean arbitrary units of fluorescence ± SD were quantified using NIS-Elements software (Nikon). ROIs were manually defined around the cell boundary and mean fluorescence intensity was measured. 

### 2.10. Immunoblotting

After washing with cold PBS, cells were lysed using the mammalian protein extraction buffer (VWR) containing protease inhibitors. The lysate was centrifuged for 15 min at 10,000× *g* at 4 °C. The resulting supernatant was transferred to a new tube and protein content was measured using a Bradford assay. Total protein (25 µg) was resolved by SDS-PAGE and transferred to nitrocellulose membrane. The membrane was blocked in TBST (50 mM Tris-Cl, 150 mM NaCl, and 0.1% Tween 20; pH 7.6) containing 5% skim milk for 45 min at room temperature. The membrane was then incubated with primary antibodies overnight. After three washes with TBST, the membrane was incubated with HRP-conjugated secondary antibody for 2 h at room temperature and washed three times. Mouse anti-GFP (sc-9996; Santa Cruz Biotechnology) and mouse anti-NPC1 (sc-271335; Santa Cruz Biotechnology) antibodies were used at 1:200. Rabit anti-GAPDH (#5174; Cell Signaling Technology) antibody was used at 1:1000. The HRP-conjugated anti-mouse or anti-rabbit secondary antibodies (R&D systems) were used at 1:2000. The luminescent signal was developed using SuperSignal WestDura Extended Duration Substrate (Thermo Fisher Scientific). We obtained images using ChemiDoc Imaging Systems (Bio-Rad).

### 2.11. Statistical Analysis

Data were expressed as means ± standard divisions obtained from three independent experiments. The significance of differences between control and treated samples was determined by Student’s *t* test using GraphPad Prism 10 software (GraphPad, La Jolla, CA, USA). A *p*-value of 0.05 or less was considered to be statistically significant.

## 3. Results

### 3.1. Bafilomycin A1, Chloroquine, and U18666A Inhibit HIV-1 Replication

We first confirmed the effects of three lysosome-targeting compounds on HIV-1 replication in the CD4^+^ T-cell line Jurkat (Figure 1). The lysosome inhibitors tested include the V-ATPase inhibitor bafilomycin A1 [20], the lysosomotropic antimalarial drug chloroquine [13,14], and the cholesterol transport inhibitor U18666A [21]. We pretreated Jurkat cells with the drugs for 2 h and then infected with HIV-1_NL4-3_ (4 ng of p24 per 2 × 10^5^ cells) in the presence of the drugs. At 4 days post-infection, we determined viral replication by measuring the levels of HIV-1 Gag (p24) released in the culture media using a standard ELISA assay. Our data indicate that bafilomycin A1, chloroquine, and U18666A inhibit HIV-1 replication (Figure 1A) in Jurkat cells without exerting an adverse effect on cell viability (Figure 1B). We also conducted a time course experiment where the effects of bafilomycin A1 (5 nM) on HIV-1 replication were investigated over longer periods. Our data indicate that bafilomycin A1 inhibits HIV-1 replication when measured at 4, 6, and 8 days post-infection without affecting cell viability (Appendix A). Our observation of U18666A-induced HIV-1 inhibition agrees with that from a previous study [22]. Similarly, our observation of chloroquine- and bafilomycin A1-induced HIV-1 inhibition is consistent with those from previous research [15,16,17,18], although there is one study that reported different results [19]. These discrepancies may be due to the differences in cell lines, viral strains, drug treatment procedures, and infection models used in each study. 

We next determined the dose-dependent effect of U18666A, bafilomycin A1, and chloroquine on HIV-1 infection using TZM-bl cells (Figure 2). TZM-bl cells express the HIV-1 receptor/coreceptor and the firefly luciferase gene under the control of the HIV-1 LTR promoter and provide a robust readout for HIV-1 infectivity [23]. The nucleoside reverse transcriptase inhibitor zidovudine was included as a control. As expected, all the compounds tested showed antiviral effect in a dose-dependent manner. The EC_50_ value of bafilomycin A1 (3.57 nM) was lower than that of zidovudine (10.2 nM), whereas the EC_50_ value of U18666A (5.84 μM) was comparable to that of chloroquine (5.82 μM). These compounds did not affect the viability of cells at the concentrations used here when measured using the MTS assay. Collectively, our findings suggest that the lysosome inhibitors bafilomycin A1, chloroquine, and U18666A inhibit HIV-1 infection and that bafilomycin A1 exerts a potent antiviral activity at a nano molar concentration range.

### 3.2. Bafilomycin A1 Inhibition of HIV-1 Is Independent of Coreceptor Tropism

Next, we determined whether the antiviral activity of bafilomycin A1 is linked to the use of a particular coreceptor by different HIV-1 strains. To test this possibility, we produced three types of HIV-1 in HEK293T cells by transfecting full-length infectious HIV-1 clones. The CXCR4-tropic virus (NL4.3) was produced using pNL4-3 [24]. The CCR5-tropic virus (NL4.3-BaL) was produced using p81A-4 [25,26,27,28], a full-length clone containing the V1–V3 envelope regions of the CCR5-tropic strain BaL in the NL4-3 background. The dual-tropic virus (89.6) capable of using both CXCR4 and CCR5 was produced using p89.6 [29,30,31]. We then tested the impact of bafilomycin A1 treatment on infections with the three HIV-1 strains in TZM-bl cells. Our data show that bafilomycin A1 efficiently inhibits infections with the viral strains with different coreceptor tropisms (Figure 3). Taken together, our findings suggest that bafilomycin A1 inhibition of HIV-1 is independent of coreceptor tropism. 

### 3.3. The Antiviral Mechanism of Bafilomycin A1 Is Different from Those of T20 and Zidovudine

To understand the mechanism of antiviral activity of bafilomycin A1, we conducted a time-of-addition assay where the test compound was added to cells at different times post-infection (Figure 4). The entry inhibitor T20, the reverse transcriptase inhibitor zidovudine, and the integrase inhibitor raltegravir were included for comparison. Bafilomycin A1 inhibited HIV-1 infection when it was added at the same time as virus (0 h) and it maintained its antiviral activity even when the compound was added later, up to 9 h post-infection (Figure 4A). The entry inhibitor T20 inhibited HIV-1 infection when the drug was added at the same time as the virus (0 h), but it lost its antiviral activity when it was added later post-infection (Figure 4B). The reverse transcriptase inhibitor zidovudine inhibited HIV-1 infection when it was added early post-infection, but its antiviral activity was gradually diminished when the drug was added later post-infection (Figure 4C). The integrase inhibitor raltegravir inhibited HIV-1 infection when it was added at the same time as the virus (0 h) and it maintained its antiviral activity even when the compound was added at late times post-infection, up to 9 h post-infection (Figure 4D). As the timing of the antiviral effect of bafilomycin A1 was different from those of T20 and zidovudine, it is likely that bafilomycin A1 inhibits HIV-1 at the step(s) of the viral life cycle other than the cell entry and reverse transcription.

### 3.4. Bafilomycin A1 Inhibits the Late Steps of HIV-1 Infection

We determined whether bafilomycin A1 interferes with the early steps of HIV-1 infection including reverse transcription, nuclear import, and integration in Jurkat cells infected with HIV-1_NL4-3_ using a qPCR analysis (Figure 5A–C). The treatment with bafilomycin did not exert a significant impact on the levels of reverse transcription products, 2-LTR circle (a marker of nuclear import), or integration products. We then investigated whether bafilomycin A1 interferes with the late steps of HIV-1 infection. To test this possibility, we transfected HEK293T cells with the proviral clone pNL4-3 in the presence of bafilomycin A1 and measured the production of virus particles. The treatment with bafilomycin A1 (5 nM) caused a significant reduction in the levels of HIV-1 Gag (p24) released in the culture media (Figure 5E) without affecting the levels of HIV-1 Gag (p24) in cells (Figure 5D). These results suggest that bafilomycin A1 may interfere with one of the late steps of the HIV-1 life cycle such as virus release.

### 3.5. Bafilomycin A1 Causes an Expansion of Lysosomes

Bafilomycin A1 is a specific inhibitor of the V-ATPase, an ATP-dependent proton pump, that is present in eukaryotic cellular membranes including the endosomes and lysosomes [20]. Given its ability to target the V-ATPase in lysosomes, we anticipated that treatment of cells with bafilomycin A1 may affect lysosome structure and/or functions. To address this possibility, we examined the changes in cellular organelles that are induced upon bafilomycin A1 treatment. We treated TZM-bl cells with bafilomycin A1 for 48 h. We then stained cells with antibodies specific for lysosomes, mitochondria, the endoplasmic reticulum, and the Golgi apparatus, and analyzed cells using confocal microscopy (Figure 6). Interestingly, the treatment with bafilomycin A1 caused an increase in the intensity of lysosomes but did not exert a significant impact on the intensities of mitochondria, the endoplasmic reticulum, and the Golgi apparatus. These results suggest that bafilomycin A1 treatment causes an increase in the number and/or size of lysosomes. These alterations in lysosome structures may reflect cellular responses to the stresses that arise under conditions of bafilomycin A1 treatment. 

### 3.6. Bafilomycin A1 Causes an Accumulation of Cholesterol in Lysosomes

As lysosomes play a critical role in cholesterol transport across the lysosomal compartments, we tested the possibility that bafilomycin A1 treatment may affect intracellular cholesterol distribution (Figure 7). To achieve this, we treated TZM-bl cells with bafilomycin A1 for 48 h. We then stained cells with filipin and an antibody that is specific for the lysosomal membrane protein LAMP1 and analyzed cells using confocal microscopy. Filipin staining is a widely accepted tool for detecting unesterified free cholesterol that accumulates in the lysosomes [32]. The treatment with bafilomycin A1 caused an accumulation of cholesterol, compared with the DMSO treatment control (Figure 7A-left,B). These results are indicative of the inhibition of lysosomal cholesterol trafficking. Our results are consistent with findings from other studies [33,34]. The treatment with bafilomycin A1 also increased the intensity of the LAMP1-positive structures, compared to the DMSO treatment control (Figure 7A-middle). Importantly, bafilomycin A1 treatment led to a large increase in the formation of filipin-positive structures colocalizing with the LAMP1-positive structures, compared to the DMSO treatment control (Figure 7A-right,C). A similar pattern of cholesterol accumulation in the lysosomal compartments was observed upon bafilomycin A1 treatment in the context of HIV-1 infection (Appendix A). Taken together, these results suggest that bafilomycin A1 interferes with cholesterol trafficking out of the lysosomes, leading to the accumulation of cholesterol in lysosomes. 

### 3.7. Overexpression of the Lysosomal Cholesterol Transporter NPC1 Partially Relieves Bafilomycin A1 Inhibition of HIV-1 Infection

As bafilomycin A1 caused the accumulation of cholesterol in lysosomes (Figure 7), we determined whether the blockage of lysosomal cholesterol trafficking is linked to the inhibition of HIV-1 infection under conditions of bafilomycin A1 treatment. Niemann–Pick type C 1 (NPC1) is a lysosomal membrane protein that is responsible for the export of unesterified cholesterol from the lysosomes [35]. We transiently expressed NPC1-GFP in TZM-bl cells and then tested the impact NPC1-GFP expression on HIV-1 infection under conditions of bafilomycin A1 treatment. As expected from the role of NPC1 in cholesterol transport, NPC1-GFP expression (Figure 8A) caused a reduction in intracellular cholesterol accumulation (Figure 8B) under conditions of bafilomycin A1 treatment. Furthermore, overexpression of GFP-NPC1 partially relieved bafilomycin A1 inhibition of HIV-1 infection (Figure 8C). Collectively, these results suggest that cholesterol trafficking across the lysosomal compartments plays a role for HIV-1 infection. 

## 4. Discussion

Lysosome inhibitors have been used as a valuable tool for dissecting the intracellular routes during infections with many viruses, in particular for those that utilize the endocytic pathway as a cell entry route. In this study, we explored the impact of three lysosome-targeting compounds (bafilomycin A1, chloroquine, and U18666A) on HIV-1 infection. Among the compounds tested, the V-ATPase inhibitor bafilomycin A1 showed the most potent inhibition of HIV-1 replication. Bafilomycin A1 inhibition of HIV-1 was independent of coreceptor tropism. A time-of-addition experiment indicated that bafilomycin A1 may interferes with HIV-1 infection at the late stage of the virus life cycle, probably following the viral genome integration in the nucleus; these include viral RNA transcription, translation, protein trafficking, and virus assembly and release. 

Interestingly, our data showed that bafilomycin A1 treatment causes a modest reduction in virus release in HEK293T cells following transfection with the HIV-1 proviral plasmid pNL4-3 (Figure 5E). However, the bafilomycin effect in Figure 5E is smaller than that in Figure 1 and Figure 8. One possible explanation for this difference may be that HEK293T cells may be less sensitive to bafilomycin compared to Jurkat or TZM-bl cells. An alternative and more plausible explanation for this difference may be that the results in Figure 1 and Figure 8 reflect the bafilomycin effect on virus replication following an infection with a replication-competent virus, whereas the results in Figure 5 reflect the bafilomycin effect on part of the late steps of virus life cycle as HEK293T cells were transfected with the proviral plasmid pNL4-3 and subsequently treated with bafilomycin. Further work is necessary to fully understand this difference.

Previous studies demonstrated an inhibitory effect of lysosome- or autophagy-targeting agents such as chloroquine and bafilomycin A1 on HIV-1 replication [15,16,17,18,36]. Our data showed an inhibition of HIV-1 replication by bafilomycin A1, whereas Garcia and colleagues demonstrated an increase in HIV-1 infectivity by the compound (**19**). Differences in experimental design may have contributed to the difference in the bafilomycin A1 effect between our study and studies by Garcia and colleagues. In our study, cells were exposed to bafilomycin A1 at 5 nM throughout the entire culture period for 48 h (TZM-bl cells) or 4 days (Jurkat cells). However, in studies by Garcia and colleagues, HeLa Magi indicator cells were treated with bafilomycin A1 at 100 nM for 16 h during infection and subsequently cells were washed to remove the drug and cultured in fresh media in the absence of the drug during the remaining 24 h. It is likely that differences in the concentrations of bafilomycin used (5 nM vs. 100 nM) and/or the time frame of drug exposure (an entire period vs. an early part of the experiment) may have resulted in the discrepancies in bafilomycin effect. 

We investigated the cellular pathways or functions that may be linked to bafilomycin A1 inhibition of HIV-1. Bafilomycin A1 treatment caused an expansion of the lysosomal compartments and an accumulation of cholesterol in lysosomes. Given the role of lysosomes in cellular cholesterol distribution [7] and the need of cholesterol for HIV-1 life cycle [3,4,5,6], it is possible that bafilomycin A1 can inhibit HIV-1 replication via disrupting cholesterol trafficking and distribution. In support of this notion, overexpression of the lysosomal cholesterol transporter NPC1 partially relieved bafilomycin A1 inhibition of HIV-1 infection. The reason that NPC1 overexpression exerts only a partial relief of bafilomycin A1 inhibition of HIV-1 is not clear. There are several possible explanations for the partial relief of bafilomycin A1 inhibition of HIV-1 by NPC1 overexpression. For example, our unpublished data suggest that bafilomycin A1 treatment causes an alteration in the autophagic flux (Appendix A). It is therefore possible that lysosome functions other than cholesterol trafficking may also play a critical role in HIV-1 replication.

The bafilomycin effect on lysosome functions has been known for a long time [37]. The major difference between our studies and other studies is that, in our studies, cells were exposed to low concentrations of bafilomycin A1 (5 nM) for the entire period of experiment (48 h or 4 days), whereas, in other studies, cells were treated with high concentrations of bafilomycin A1 (0.1 to 1 µM) for 2 to 3 h and subsequently cells were washed and cultured in the absence of the compound. Therefore, our studies may more closely reflect the physiological conditions, compared with the studies that have used washout experiments. A complex interaction between autophagy and HIV-1 has been reported. For the way HIV-1 and autophagy influence each other, discordant results have been reported in studies using different in vitro models. Depending on the cell type and on the infectious status of the target cell type, different effects of HIV-1 on autophagy have been mentioned [38,39,40,41,42,43,44,45,46]. Similarly, different outcomes have been reported in how autophagy modulates HIV-1 replication [43,46,47,48]. Collectively, these findings suggest that the discordance between different studies could be due to the different cellular models used and the differences in the cellular status. A recent study showed that modulation of the autophagic pathway inhibits HIV-1 infection in human lymphoid tissue cultured ex vivo, suggesting that HIV-1 replication requires a fine-tuned level of autophagy [18]. As autophagic flux requires intact intracellular cholesterol trafficking [34], it is likely that bafilomycin A1-induced cholesterol trafficking defects cause a disruption in autophagy, which in turn affects HIV-1 replication. 

In summary, we have demonstrated that the lysosome-targeting compound bafilomycin A1 inhibits HIV-1 replication at least in part via disrupting the lysosomal cholesterol trafficking. It is important to identify the precise steps of HIV-1 replication affected by bafilomycin A1 treatment and to understand the changes in cellular pathways responsible for HIV-1 inhibition. Further research is necessary to define a direct link between HIV replication and lysosome size, positioning, or cholesterol accumulation. The outcomes of these studies may provide important new insights into HIV biology and can lead to improved therapeutic approaches for HIV/AIDS.

## Figures and Tables

**Figure 1 viruses-16-01374-f001:**
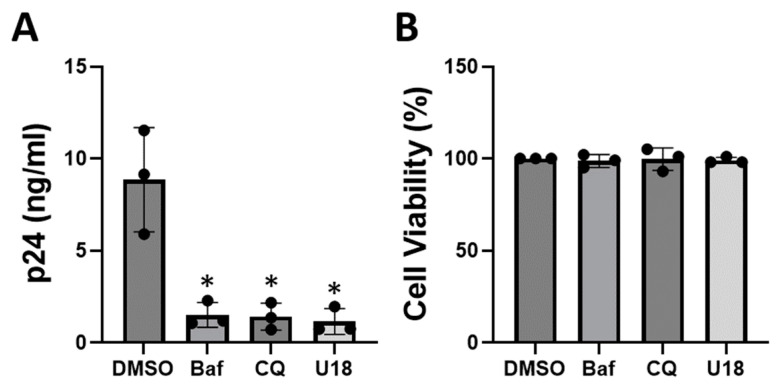
Bafilomycin A1, chloroquine, and U18666A inhibit HIV-1 replication in Jurkat cells. For virus replication (**A**), cells were pretreated with the drugs for 2 h and then infected with HIV-1_NL4.3_ (4 ng of p24/2 × 10^5^ cells) in the presence of the drugs. The levels of HIV-1 Gag (p24) released in the media were determined using a standard ELISA assay at 4 days post-infection. Bafilomycin A1 (Baf), 5 nM. Chloroquine (CQ), 20 μM. U18666A (U18), 20 µM. DMSO, 0.1% (*v*/*v*). Cell viability (**B**) was measured using the MTS assay (Promega) after treating Jurkat cells with the drugs for 48 h. Values relative to the DMSO control (100%) are shown. Data are means ± SD (*n* = 3). * *p* < 0.05.

**Figure 2 viruses-16-01374-f002:**
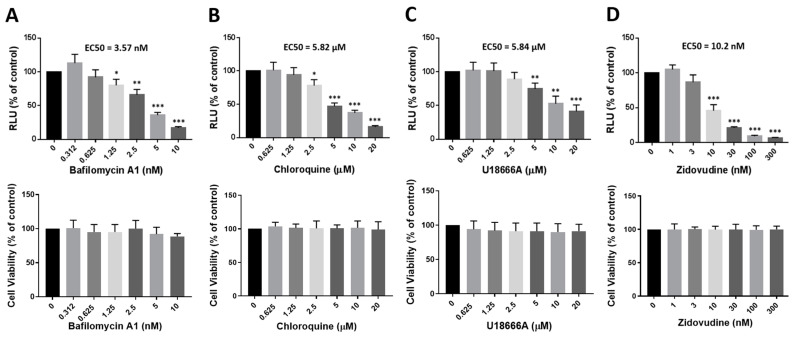
Dose-dependent inhibition of HIV-1 by bafilomycin A1, chloroquine, and U18666A. TZM-bl cells were pretreated with bafilomycin A1 (**A**), chloroquine (**B**), U18666A (**C**), or zidovudine (**D**) at the concentrations indicated for 2 h and then infected with HIV-1_NL4.3_ (2 ng of p24/2 × 10^5^ cells) in the presence of the drugs. At 48 h post-infection, infectivity was determined using a luciferase assay (Promega); the relative light unit (RLU) relative to the DMSO control (100%) is shown. Cell viability was measured using an MTS assay (Promega); the cell viability relative to the DMSO control (100%) is shown. EC_50_ values were calculated by a nonlinear regression analysis using GraphPad Prism. Data are means ± SD (*n* = 3). * *p* < 0.05; ** *p* < 0.01; *** *p* < 0.001.

**Figure 3 viruses-16-01374-f003:**
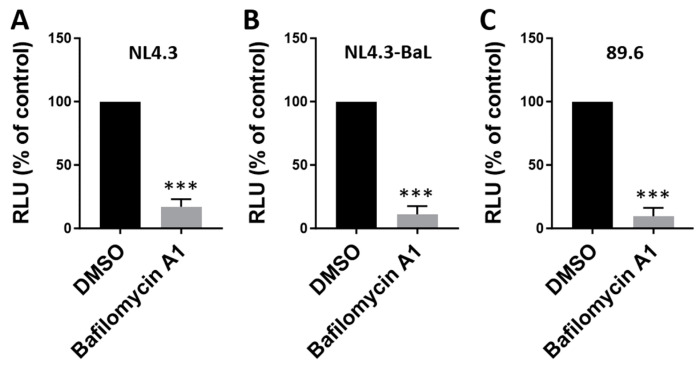
Bafilomycin A1 inhibition of HIV-1 is independent of coreceptor tropism. TZM-bl cells were infected with CXCR4 tropic HIV-1_NL4.3_ (**A**), CCR5 tropic HIV-1_NL4.3-BaL_ (**B**), or dual tropic HIV-1_89.6_ (**C**) (2 ng of p24/2 × 10^5^ cells) in the absence or presence of bafilomycin A1 (5 nM). At 48 h post-infection, infectivity was measured using a luciferase assay. Values relative to the DMSO control (100%) are shown. RLU; relative light unit. Data are means ± SD (*n* = 3). *** *p* < 0.001.

**Figure 4 viruses-16-01374-f004:**
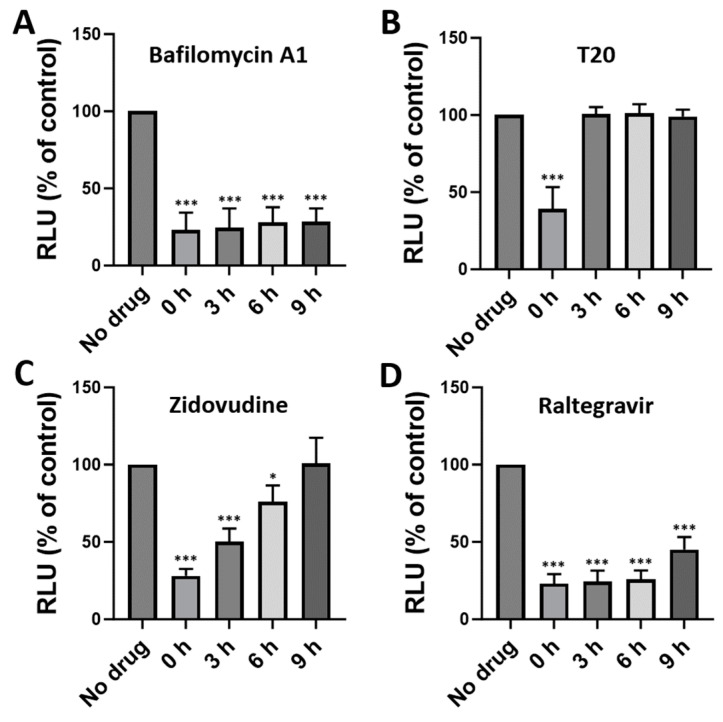
The antiviral activities of bafilomycin A1 (**A**), T20 (**B**), zidovudine (**C**), and raltegravir (**D**) in a time-of-addition assay. TZM-bl cells were infected with HIV-1_NL4.3_ (2 ng of p24/2 × 10^5^ cells) with the drugs added at the indicated times. At 48 h post-infection, infectivity was measured using a luciferase assay. Bafilomycin A1, 5 nM; T20, 0.5 μM; zidovudine, 0.5 μM; raltegravir, 0.5 μM. RLU; relative light unit. Data are means ± SD (*n* = 3). * *p* < 0.05; *** *p* < 0.001.

**Figure 5 viruses-16-01374-f005:**
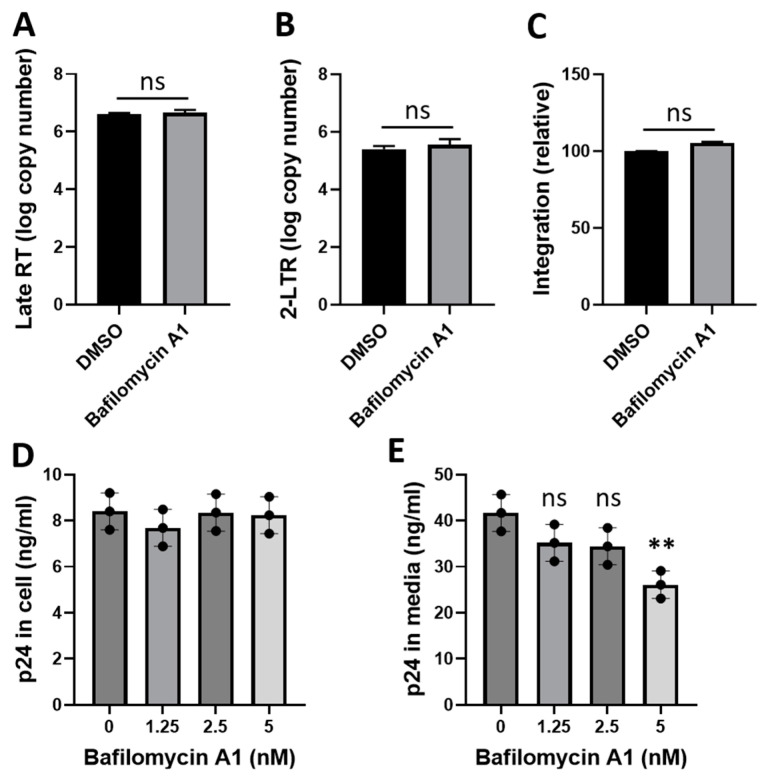
Effects of bafilomycin A1 on reverse transcription, nuclear import, integration, and virus particle production. (**A**–**C**) Jurkat cells were pretreated with bafilomycin A1 (5 nM) for 2 h and then infected with HIV-1_NL4.3_ (4 ng of p24/2 × 10^5^ cells) in the presence of bafilomycin A1 (5 nM). At 24 h post-infection, total DNAs were isolated and the levels of late reverse transcription (RT) products (**A**), 2-LTR circles (**B**), or integration products (**C**) were determined using qPCR analysis. (**D**,**E**) HEK293T cells were transfected with the proviral molecular clone pNL4-3 plasmid. At 4 h post-transfection, cells were washed and incubated in fresh media in the absence or presence of bafilomycin A1. At 28 h post-transfection, the levels of HIV-1 Gag (p24) in cells (**D**) and in the culture media (**E**) were determined using a standard ELISA assay. Data are means ± SD (*n* = 3). ** *p* < 0.01. ns, non-significant.

**Figure 6 viruses-16-01374-f006:**
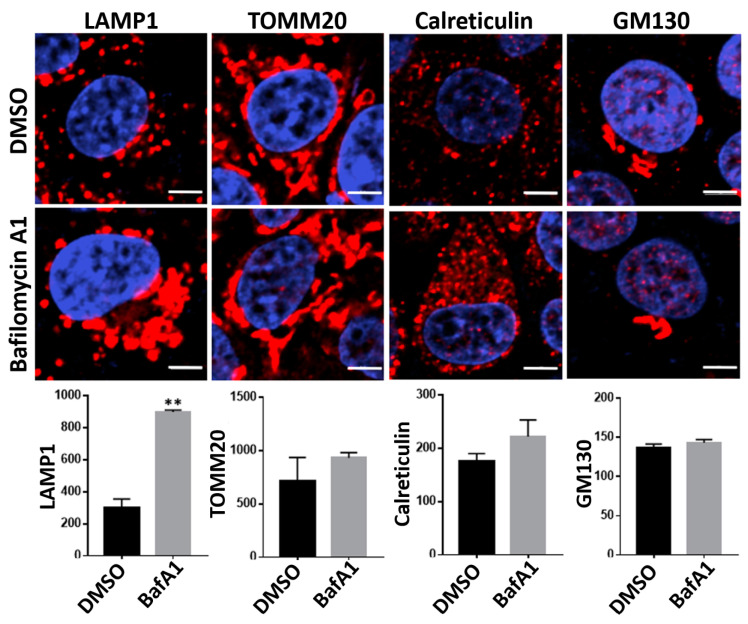
Bafilomycin A1 causes an expansion of lysosomes. TZM-bl cells were treated with bafilomycin A1 (5 nM) or DMSO for 48 h. Cells were stained using antibodies specific for LAMP1 (lysosome marker), TOMM20 (mitochondrial marker), calreticulin (ER marker), and GM130 (Golgi marker) in combination with CF594-conjugated secondary antibody (Red). DAPI (blue) was used to stain nuclei. Images (60×) were obtained using the Nikon A1R confocal microscope. Scale bar = 5 µm. The images are representative of three independent experiments. The bar graphs below the images show mean fluorescence intensity from 30 cells in 4 different fields. Data are means ± SD (*n* = 3). ** *p* < 0.01.

**Figure 7 viruses-16-01374-f007:**
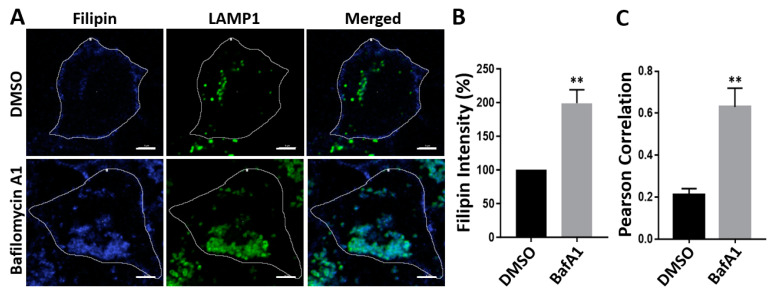
Bafilomycin A1 causes an accumulation of cholesterol in lysosomes. TZM-bl cells were treated with DMSO or bafilomycin A1 (5 nM) for 48 h. Unesterified free cholesterol was detected using filipin. Lysosomes were detected using anti-LAMP1 antibody with CF488-conjugated secondary antibody. Images (**A**) (60×) were obtained using the Nikon A1R confocal microscope. Scale bar = 5 µm. Images are representative of three independent experiments. Filipin intensity (**B**) and Pearson’s correlation coefficient (**C**) of filipin and LAMP1 signals were measured from 60 cells in 4 different fields. ** *p* < 0.01.

**Figure 8 viruses-16-01374-f008:**
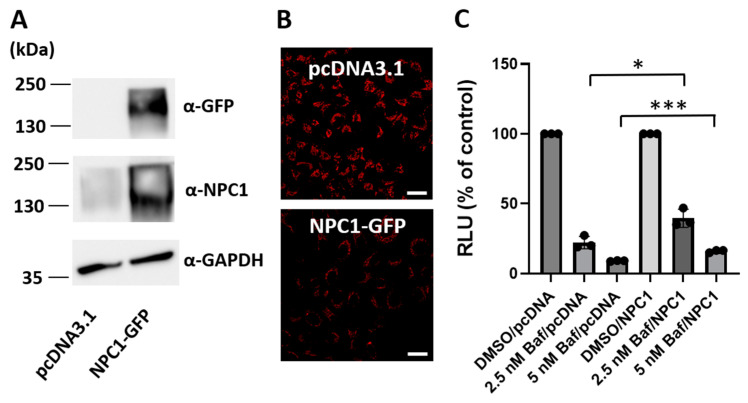
NPC1 overexpression partially relieves bafilomycin A1 inhibition of HIV-1 infection. (**A**) TZM-bl cells were transfected for 48 h with a plasmid expressing NPC1-GFP or the empty control plasmid pcDNA3.1. Western blotting was conducted to confirm the expression of NPC1-GFP using an antibody specific for GFP or NPC1. GAPDH was included as a loading control. (**B**) Four hours after transfection with the NPC1-GFP expressing plasmid or the empty vector control, cells were treated with bafilomycin A1 (5 nM) for 48 h and free cholesterol was detected using filipin. Images (60×) were obtained using the Nikon A1R confocal microscope. Scale bar = 20 µm. Images are representative of three independent experiments. (**C**) At 24 h post-transfection with the NPC1-GFP expressing plasmid or the empty vector control, cells were infected with HIV-1 in the absence or presence of bafilomycin A1 (5 nM). At 48 h post-infection, HIV-1 infection was determined using a luciferase assay. Data are means ± SD (*n* = 3). * *p* < 0.05. *** *p* < 0.001.

## Data Availability

All the data supporting this study are included in the article. Further inquiries should be directed to the corresponding author.

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
