# Peer review of "Bafilomycin A1 Inhibits HIV-1 Infection by Disrupting Lysosomal Cholesterol Transport"

_viruses, 2024, doi:10.3390/v16091374_

Round 1

Reviewer 1 Report

Comments and Suggestions for Authors

The authors investigated the effects of lysosomal inhibitors bafilomycin A1, chloroquine, and U18666A on HIV infection and replication. They describe the effects of bafilomycin in detail reporting an inhibition in HIV replication. In Jurkat cells the drug is inhibiting late steps in virus replication. The data are interesting, well described,  and shed new light on late steps in HIV replication.

The authors pretreated cells with bafilomycin A1 and measured HIV production in media after 4 days, and noted  a strong inhibition of virus production, and performed a drug titration assay to estimated the EC50 of the drug in Jurkat cells. These data are clear and well described. The authors have used only a single time point (4 days post infection) to measure HIV production. This limited experiment is insufficient to describe the effect. Bafilomycin is reported to have pleiotropic effects in tissue culture systems, which may have effects on HIV replication in addition to autophagy. Specifically, it is not known whether bafilomycin causes inhibition of replication or simply a delay in replication.  The authors need to perform time course studies with frequent sampling over longer periods to characterize the effect of bafilomycin in HIV infection.

Similarly, toxicity studies were performed that quantified cell death at 48 h exposure. No studies were performed to estimate toxicity of the drug after 4 days, (the timing of the HIV replication experiments). The authors need to perform more extensive toxicity studies beyond 48 h to determine whether toxic effects of bafilomycin over longer periods may contribute to the decreases in HIV production. For instance Biard-Piechaczyk reported HIV can induce autophagy by binding of env to CxCR4, resulting in bystander death  if blocking autophagy decreases HIV production, does it decrease numbers of HIV un-infected cells after 4 days?

The authors need to explain the variability in the magnitude of the bafilomycin response in TZM-bl cells reported in figures 2 and 3. In figure 2, the effect of 5 nM bafilomycin on HIV production appears to be less than 3 fold, but in figure 3 the effect seems closer to ten-fold.

The authors need to justify the need for Figure 5 panel D. 5D shows the effect of bafilomycin on HIV particle production in HEK293T cells, but the data are somewhat confusing when compared with virus production data in Figure 1and in Figure 8. The bafilomycin effect in 5D, though statistically significant, is relatively modest, perhaps less than 2-fold. The data in Figure 1 with Jurkat cells, or figure 8C is more compelling (5-10 fold). Are the authors suggesting that HEK293T cells are less sensitive to  bafilomycin? These findings with HEK293T are not discussed in light of the other experiments, and its not clear whether this experiment adds or detracts from the paper; for clarity sake, 5D should be removed.

At several points data are presented which appear to be confirmatory of previous studies: 1) the authors describe that bafilomycin causes expansion of lysosomes, which has been reported for a number of years (Yashimori JBC 1991; Yamoto et al. 2019). 2) the effects of bafilomycin A1 on cholesterol transport using fillipin staining have been previously described (e.g., Maharajan et al., 2022).  How are the data presented in this submission different from prior studies? In addition, no direct link between HIV replication and lysosome size, or cholesterol accumulation is presented. Bafilomycin has many effects on cells, which have not been investigated in this system;  the observations presented, though interesting, may be irrelevant to HIV replication.  If largely confirmatory, the data can be moved to the supplement.

Previously, Garcia and coworkers and Rekosh and coworkers demonstrated increase in infectivity with bafilomycin A1 in  monocytic cell lines. Here the authors rule the possibility that bafilomycin A1 inhibits early events in replication using time addition experiments. The authors should discuss why their findings are different from previous studies.

Minor:

 line 463 “whicn” should read “which”

 line 396- double period

Author Response

Comment 1: The authors have used only a single time point (4 days post infection) to measure HIV production. This limited experiment is insufficient to describe the effect. Bafilomycin is reported to have pleiotropic effects in tissue culture systems---. The authors need to perform time course studies with frequent sampling over longer periods to characterize the effect of bafilomycin in HIV infection.

Response to Comment 1: We agree with this comment. We conducted a time course experiment, and our data indicate that bafilomycin A1 (5 nM) inhibits HIV-1 replication when measured at 4, 6, and 8 days post-infection without affecting cell viability.

This response is described in the revised manuscript: Results 3.1 (page 6, lines 249-252).

The supporting data can be found in Supplementary Information: Figure S1.

Comment 2: The authors need to perform more extensive toxicity studies beyond 48 h to determine whether toxic effects of bafilomycin over longer periods may contribute to the decreases in HIV production.

Response to Comment 2: Thank you for pointing this out. As described in the previous response, bafilomycin A1 (5 nM) did not exert an adverse effect on cell viability over 8 days after the treatment.

This response is described in the revised manuscript: Results 3.1 (page 6, lines 249-252).

The supporting data can be found in Supplementary Information: Figure S1.   

Comment 3: The authors need to explain the variability in the magnitude of the bafilomycin response in TZM-bl cells reported in figures 2 and 3. In figure 2, the effect of 5 nM bafilomycin on HIV production appears to be less than 3 fold, but in figure 3 the effect seems closer to ten-fold.

Response to Comment 3: We agree with this comment. Indeed, our data show a variability in the magnitude of bafilomycin effect on HIV production. One possibility is that the aliquots of bafilomycin solution used in this study may show a variability in anti-HIV activity after multiple freeze-thaw cycles.  

Comment 4: The authors need to justify the need for Figure 5 panel D. 5D shows the effect of bafilomycin on HIV particle production in HEK293T cells, but the data are somewhat confusing when compared with virus production data in Figure 1and in Figure 8. The bafilomycin effect in 5D, though statistically significant, is relatively modest, perhaps less than 2-fold. The data in Figure 1 with Jurkat cells, or figure 8C is more compelling (5-10 fold).

Response to Comment 4: We agree with this comment. Indeed, the bafilomycin effect in 5D (5E in the revised manuscript) is smaller than that in Figure 1 and Figure 8. One possible explanation for this difference may be that HEK293T cells may be less sensitive to bafilomycin. An alternative and more plausible explanation for this difference may be that the results in Figure 1 and Figure 8 reflect the bafilomycin effect on virus replication following an infection with a replication-competent virus, whereas the results in Figure 5D (5E in the revised manuscript) reflect the bafilomycin effect on some of the late steps of virus replication as HEK293T cells were transfected with the proviral plasmid pNL4-3 and subsequently treated with bafilomycin.

This response is described in the revised manuscript: Discussion (page 12, lines 446-454).

Comment 5: How are the data presented in this submission different from prior studies showing the bafilomycin effect on lysosomes?

Response to Comment 5: Thank you for pointing this out. As the reviewer mentioned, the bafilomycin effect on lysosome functions has been known for a long time. The major difference between our studies and the majority of other studies is that in our studies cells were exposed to low concentrations of bafilomycin A1 (5 nM) for the entire period of experiment (48 hours or 4 days), whereas in other studies cells were treated with high concentrations of bafilomycin A1 (0.1 to 1 µM) for 2 to 3 hours and subsequently cells were washed and cultured in the absence of the compound.

This response is described in the revised manuscript: Discussion (page 13, lines 484-489).

Comment 6: Previously, Garcia and coworkers demonstrated an increase in infectivity with bafilomycin A1. Here the authors rule the possibility that bafilomycin A1 inhibits early events in replication using time-of-addition experiments. The authors should discuss why their findings are different from previous studies.

Response to Comment 6: Thank you for pointing this out. Our data showed an inhibition of HIV replication by bafilomycin A1 whereas Garcia and colleagues demonstrated an increase of HIV infectivity by the compound. Differences in experimental design may have contributed to the difference in the bafilomycin A1 effect between our study and studies by Garcia and colleagues. In our study, cells were exposed to bafilomycin A1 at 5 nM throughout the entire period for 48 hours (TZM-bl cells) or 4 days (Jurkat cells). However, in studies by Garcia and colleagues, HeLa Magi indicator cells were treated with bafilomycin A1 at 100 nM for 16 hours during infection and subsequently cells were washed to remove the drug and cultured in fresh media in the absence of the drug during the remaining 24 hours. It is likely that differences in the concentrations of bafilomycin used (5 nM vs. 100 nM) and/or the time frame of drug exposure (an entire period vs. an early part of the experiment) may have resulted in the discrepancies in bafilomycin effect.

This response is described in the revised manuscript: Discussion (page 12, lines 455-468).

Comment 7:  line 396- double period. line 463 “whicn” should read “which”.

Response to Comment 7: Thank you for finding out these errors. These errors have been corrected in the revised manuscript.

Reviewer 2 Report

Comments and Suggestions for Authors

Very nice written and important study! No major comments.

Author Response

Thank you for the kind comments.

Reviewer 3 Report

Comments and Suggestions for Authors

The study examines the effect of Bafilomycin A1 (BafA1) on HIV-1 infectivity. While the effect of BafA1 is well-documented in published literature, its mechanism of action remains unclear. In this study, the authors demonstrate a dose-dependent inhibitory effect of BafA1 on HIV-1. Including a positive control for the inhibitory activity of Bafilomycin A1 (BafA1) on autophagy in the cell lines used in the study is essential. By using inhibitors that target different stages of HIV-1 infection, they conclude that BafA1 inhibits HIV-1 at a later stage in the virus life cycle, possibly during assembly or egress. However, this conclusion lacks experimental evidence. Measuring intracellular levels of Gag to assess virus release efficiency (Figure 5D) is crucial. Additionally, the authors suggest that BafA1's inhibitory effect on HIV-1 may be related to its inhibition of lysosomal cholesterol trafficking. Yet, they do not present data on cholesterol status in BafA1-treated cells in the context of HIV-1 infection. The rescue experiment involving the overexpression of the cholesterol-trafficking protein NPC1 is not convincing as the rescue is modest, and the colocalization of cholesterol with lysosome marker in the context of HIV-1 infection is not addressed. I believe this study is preliminary and not suitable for publication at this stage.

Author Response

Comment 1: They conclude that BafA1 inhibits HIV-1 at a later stage in the virus life cycle, possibly during assembly or egress. However, this conclusion lacks experimental evidence. Measuring intracellular levels of Gag to assess virus release efficiency (Figure 5D) is crucial.

Response to Comment 1: We agree with this comment. Therefore, we performed the experiment and measured the levels of HIV-1 Gag (p24) in cells and in the culture media.

This response is described in the revised manuscript: Results 3.4 (page 9, lines 339-343 and 350-355).

These new results can be found in Figure 5D (p24 in cell) and Figure 5E (p24 in media).

Comment 2: The authors suggest that BafA1's inhibitory effect on HIV-1 may be related to its inhibition of lysosomal cholesterol trafficking. Yet, they do not present data on cholesterol status in BafA1-treated cells in the context of HIV-1 infection.

Response to Comment 2: Thank you for pointing this out. We have data showing a similar pattern of cholesterol accumulation in the lysosomal compartments upon bafilomycin A1 treatment in the context of HIV-1 infection.

This response is described in the revised manuscript: Results 3.6 (page 10, lines 394-396).

These results can be found in Supplementary Information: Figure S2.  

Comment 3: The rescue experiment involving the overexpression of the cholesterol-trafficking protein NPC1 is not convincing as the rescue is modest.

Response to Comment 3: We agree with this comment. NPC1 overexpression results in only a partial rescue of bafilomycin effect on HIV replication. We have indirect evidence that lysosome functions other than cholesterol trafficking may play a role in HIV replication.

This response is described in the revised manuscript: Discussion (pages 12-13, lines 477-483).

Reviewer 4 Report

Comments and Suggestions for Authors

Dear Authors,

The manuscript reports on bafilomycin A1 with nanomolar inhibitory effect on HIV infection.

The manuscript contains description of various experiments related to inhibition of HIV replication, its effect on the late steps of HIV life cycle. 

At the same time, the activity of bafilomycin A1 and some other autophagy-enhancing drugs on HIV replication and their possible interference with HIV infection has been described previously. Therefore, the novelty of the results presented in the current manuscript needs to be carefully identified and described. The studies that have previously evaluated the activity of bafilomycin A1 should be cited and discussed in comparison with the results of the current study.

Late steps of HIV infection are rather late steps of the HIV life cycle in the host cell. Late steps of HIV infection are more likely to be considered AIDS.

Comments on the Quality of English Language

Minor changes of English language are required.

Author Response

Comment 1: The activity of bafilomycin A1 and some other autophagy-enhancing drugs on HIV replication has been described previously. Therefore, the novelty of the results presented in the current manuscript needs to be carefully identified and described. The studies that have previously evaluated the activity of bafilomycin A1 should be cited and discussed in comparison with the results of the current study.

Response to Comment 1: Thank you for pointing this out. According to the reviewer’s suggestion, we have cited the previous studies that described the effects of lysosome- and autophagy-targeting compounds such as chloroquine and bafilomycin A1 in the revised manuscript. We also discussed the novelty of our results in comparison with the results of the studies from other groups.

This response is described in the revised manuscript: Discussion (lines 455-468, page 12).

Round 2

Reviewer 4 Report

Comments and Suggestions for Authors

Dear Authors,

thank you very much for corrections according to the comments of the Review (the 1st round).

I have minor comment:

The phrase: "the major difference between our studies and the majority of other studies is that in our studies cells were exposed to low concentrations of bafilomycin A1 (5 nM) for the entire period of experiment (48 hours or 4 days), whereas in other studies cells were treated with high concentrations of bafilomycin A1 (0.1 to 1 µM) for 2 to 3 hours and subsequently cells were washed and cultured in the absence of the compound."

Please, correct the repeated term: "major" and "majority" in one sentence.

Also, this statement lacks the conclusion that will reflect the results of the study appropriately. Please, add the conclusion.

Comments on the Quality of English Language

Minor corrections of English are needed.

Author Response

(The authors gave the same response as above.)
